# Directed Structural Adaptation to Overcome Statistical Conflicts and Enable Continual Learning

## Abstract

Adaptive networks today rely on overparameterized fixed topologies that cannot break through the statistical conflicts they encounter in the data they are exposed to, and are prone to "catastrophic forgetting" as the network attempts to reuse the existing structures to learn new task. We propose a structural adaptation method, DIRAD, that can complexify as needed and in a directed manner without being limited by statistical conflicts within a dataset. We then extend this method and present the PREVAL framework, designed to prevent "catastrophic forgetting" in continual learning by detection of new data and assigning encountered data to suitable models adapted to process them, without needing task labels anywhere in the workflow. We show the reliability of the DIRAD in growing a network with high performance and orders-of-magnitude simpler than fixed topology networks; and demonstrate the proof-of-concept operation of PREVAL, in which continual adaptation to new tasks is observed while being able to detect and discern previously-encountered tasks.

## 1 Introduction

Past decade has shown that complex networks should be at the core of any AI system that needs to be of robust use in any task of reasonable complexity. It has, however, been unfortunate that over the same period, the field of machine learning (ML) has been stuck in the twin limiting paradigms of static topologies and statistical fine-tuning, attempting to make up for the limitations of both of these by using brute force, in form of overparameterization and computational requirements accompanying it. Limitations imposed by these paradigms also prevent solving the crucial problem of "catastrophic forgetting" in continual learning, which becomes especially important in systems that need to learn continually without explicit storage and replay of past data. In this work, we first propose a novel method of structural adaptation, operating with gradient descent, with a strong bias towards minimal network complexity (i.e. size). Our framework, first of its kind to the best of our knowledge, is neither limited by *statistical conflicts between samples* (a term, detailed in the text, we use to refer to conflicting requirements within a dataset that result in net zero adaptive pressure on parameters, despite nonzero change requirements for any individual sample) nor reliant on excess network complexity to find a solution with strong guarantees. We apply this method to construct a system to prevent "catastrophic forgetting" by recognizing unexpected data, constructing new components (models) to process such new data without affecting past responses, and then selecting the suitable one over an existing array of such models when data from a past task is provided - a unified continual learning framework that does not require task labels or switching signals anywhere, neither during adaptation nor deployment; all while not relying on storage and replay of past samples. Finally, we provide positive results on both of these frameworks in our initial experiments on MNIST and FashionMNIST datasets.

**Note on terminology** The mechanisms we propose in this paper are not biologically plausible at neuronal level, and sharing a nonlinear weighted-sum paradigm is not enough to justify an analogy with neural systems given the additional mechanisms we introduce. To avoid implying such an analogy and to accurately describe these mechanisms, we use the terms *node* and *edge* instead of

"neuron" and "synapse". We also refrain from using the term "neural network (NN)" and simply use *network* when referring to our design.

## 2 BACKGROUND AND RELATED WORK

**Structural adaptation**   Structural adaptation in NNs hasn't gained as much attention as other aspects of this technology, as many of these methods involve an additional step and often don't provide a significant benefit compared to the added network complexity (i.e. number of nodes/edges). One subfield in literature, called "neural architecture search," focuses on optimizing the architecture itself explicitly (Liu et al. (2018); Shin et al. (2018); Baker et al. (2016); Stanley et al. (2019); Liu et al. (2017); Miikkulainen et al. (2019)). Some other works view "structural adaptation" as starting from scratch or growth, sometimes referred to as Artificial Embryogenesis (Kowaliw et al. (2014)), often using evolutionary algorithms. Recent methods for expanding neural networks, similar to our design choices, can also be found in the literature (Dai et al. (2019); Evci et al. (2022); Mitchell et al. (2023)). Our approach aligns more with the group that designs developmental methods rather than relying on external loops or added pressures for architecture optimization. We use structural adaptation not for topology optimization but to drive further response adaptation. However, our approach differs from this group in that we prioritize minimal network complexity and address statistical conflicts while operating outside the conventional NN paradigm.

**Continual adaptation**   A significant issue with continual learning or adaptation is "catastrophic forgetting" or, as we call it, "destructive adaptation"[1] (DA) - when the new instances differ significantly from previously observed examples, they cause the new information to overwrite previously learned knowledge in the network, a problem until today remains without a reliable solution (Hadsell et al. (2020); Parisi et al. (2019)). Systems with fixed capacity cannot deal with the problem adequately: An existing capacity is always used completely for the previous tasks (since information is engrained in a neural network in a distributed manner), and existing information will be eventually (often immediately) lost as new tasks differing significantly from the previous ones arrive. The methods that work by the addition of capacity, on the other hand, cannot autonomously decide when to add capacity, how to assign different added components to different tasks, and how to choose among components when presented with one of the past tasks (e.g. in (Rusu et al. (2016)) it must be externally signalled to the system both when a new task is observed, and for recall, it must be specified which of the past tasks is being seen) - the same limitation also applies to methods that explicitly store information about the solutions of past tasks as well (e.g. (Kirkpatrick et al. (2017))). Some extensions of such methods that partially try to address these questions require task labels during adaptation phase and have no mechanism that can detect new tasks (e.g. (Jacobson et al. (2022))), hence cannot constitute systems with autonomous continual adaptation capability. Other methods claim to operate without such "task boundaries", yet they all rely on explicitly storing and then replaying past data (Aljundi et al. (2019), Table 1 of Buzzega et al. (2020)) - this is infeasible in many real-world scenarios due to the lifelong memory requirements of the systems, confidentiality or data protection considerations; and obviously do away with one of the primary motivators for continual learning. We are unaware of a framework proposed against DA that does not store and replay of past data, operate without external signalling of task change, is able to recognize a new task, and add new system components as needed - in other words, a full continual learning setup without any simplifying assumptions. Designing such a method is what we do in this work.

**Novelty/anomaly detection**   Methods of novelty/anomaly detection are those that are interested in classifying certain encounters as novel or not. Detailed survey of such methods is beyond our scope, interested readers can find a review in (Pimentel et al. (2014)). The field itself is not of primary interest to us except as a sub-goal, since it concerns itself with systems designed to classify samples as novel or not, while we want to both quantify and localize this novelty, doing that within a system that is actually used for the performation of a particular task. Furthermore, as methods susceptible to statistical conflicts, they are not suitable for our purposes (as will be discussed in the following sections).

---

[1]We think "catastrophic forgetting" is unnecessarily anthropomorphized (the phenomenon is a challenge to all adaptive systems) and does not correspond to the gradual process of "forgetting" as commonly understood but to active destruction of past information.

## 3    MECHANISMS OF STRUCTURAL ADAPTATION

In this section, we describe our structural adaptation and network growth mechanism. The mechanisms described here are *not* directly aimed at the prevention of destructive adaptation (DA), but are general adaptive processes that can be used in any ML problem. Throughout the section, we discuss a single task. Here we only provide a summary of the method and its core points. The full theoretical development, with justifications of design choices and practical considerations, can be found in the Appendix. To illustrate the process in action, we provide an example path of adaptation in Figure 1, to which we refer throughout our narrative below.

We always assume a network starting with only the input and output nodes, and no *hidden nodes* (neither input nor output - Figure 1a) - however the mechanisms we designed can operate locally within networks of arbitrary standing complexity. Our aim is to develop a network that can complexify as needed, but not more (prioritizing parameter adaptation where possible); and that is not limited by statistical trade-offs between different samples in a batch. We call the processes that grow the network by introducing new components as *generative processes (GPs)*, each of which is *neutral*: No node's response is changed due to a GP; and all changes in net response occur under the influence of gradients to ensure no harm to performance. Parameter adaptation within network (edge weights and node biases) are done by standard gradient descent via backpropagation algorithm.

**Adaptive potentials (APs)**    The *immediate AP* of an edge $(i, j)$ is defined as the net gradient that this edge's weight gets over a given batch, i.e. $\partial C / \partial w_{ij}$ where $C$ is total (summed across samples) cost/error for that batch. We say that the immediate AP of edge is *exhausted* if $\partial C / \partial w_{ij} \approx 0$. Analogously, we say that the *immediate AP of a node $j$ is exhausted* if $\partial C / \partial z_j = \delta_j \approx 0$ (where $z_j$ is the activation) and the immediate AP of all its in-edges is exhausted as well. We define the *total AP* of a node as $\sum_m |\delta_j^m|$ ($m$ is the sample index), as a measure of the total adaptive gain that can be obtained by a change in the activation of that node $z_j$, if it can be exploited. Analogously, we define the *total AP of an edge (i,j)* as $\sum_m |\partial C^m / \partial w_{ij}|$.

**Edge generation**    The first GP, edge generation, generates an in-edge with an initial weight of $0$ to a node $j$. The source of the edge is chosen among the candidates to be the one that maximizes the magnitude of the expected immediate gradient update on the edge, i.e. the value $|\partial C / \partial w_{ij}| \propto \left| \sum_m a_i^m \delta_j^m \right|$ where $a_i^m$ is the state (response) of node $i$. We generate an edge for a node if the immediate AP of the node is exhausted, but its total AP is not (i.e. is nonzero), see Figure 1b. Intuitively, this operation allows us to take a node with a nonzero total AP out of the exhaustion of its immediate AP provided that there are sources that can be a good match with the change directions requested by the gradients.

**Edge-node conversion (ENC) and resolving statistical conflicts**    The ENC mechanism is designed to operate where the immediate AP of an edge is exhausted while its total AP is not. Recall that total AP quantifies the total adaptive gain that can be obtained from a given edge. This potential, no matter how large, cannot be utilized in NNs if the immediate AP of the edge is exhausted (net gradient 0). ENC mechanism provides a solution to that by modulating the gradients of the original edge (upon the progression of adaptation) so that they become aligned instead of opposing. Here we only describe the final design of the ENC operation with brief intuition on their justifications where applicable - for detailed reasoning, the reader is referred to the Appendix.

When an edge $(i, j)$ undergoes ENC, it is replaced with a new node $k$ and two edges $(i, k)$ and $(k, j)$ that become the new path connecting $i$ to $j$. The new node is *modulatory*, whose state is computed by the multiplication of two terms:

$$a_x^m = \prod_{i \in \{0,1\}} \sigma_i \left( \sum_{y \in in_i(x)} w_{yx} a_y^m + b_{x,i} \right) \tag{1}$$

where subscripts $x, i$ refer to *node x, term i*. We also assume two distinct transfer functions $\sigma_0$ and $\sigma_1$, where $\sigma_0(z) = z$ in our design and $\sigma_1$ is given as $\sigma_1(x) = 4/(1 + e^{-Kx}) - 1$, where $K = 1/w_{ij}$ is a node-specific property. This function can take values in the range $(-1, 3)$, and hence is able to invert the sign of the previously opposing gradients. Nonlinearity in network is realized by the

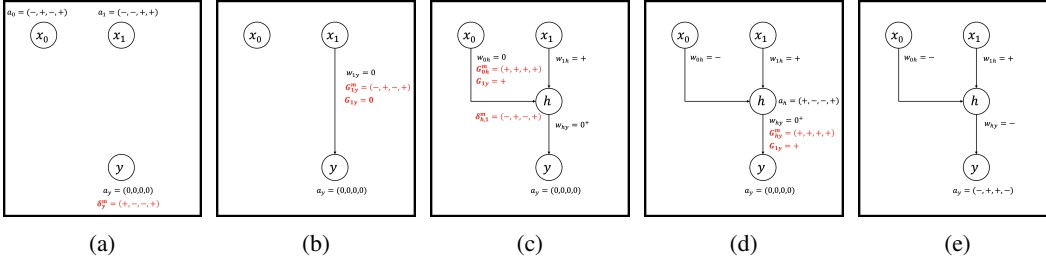

Figure 1: A simplified illustrative case of the path of adaptation for signed XOR ("False" represented by $-1$ instead of 0). Inputs: $x_0$, $x_1$. Output: $y$. In the figures, $G_e$ represent $dC/dw_e$, $a_i$ state of node $i$, and the four values in parentheses represent the signs that a variable takes for the four samples, respectively. We simplify by assuming no bias and that the adaptation process of different components happen in sequence instead of simultaneously. (a) Initial state. $y$ has immediate AP exhausted but total AP nonzero, hence will form an edge. Neither input is an immediately-useful source, since neither matches with deltas of $y$. (b) $y$ forms an in-edge, source chosen randomly. The generated edge has a net gradient of 0 and hence cannot proceed with adaptation. This is a "local optimum" in a static network. (c) The new edge undergoes ENC from exhaustion, and its gradient is transferred to the Term 1 deltas of the new node $h$. $h$, under immediate AP exhaustion, gets an edge from $x_0$ that can provide perfect match with the sign of its deltas and creates a large positive net gradient for the new edge. (d) Modulatory edge to term 1 of the converted node $h$ stabilized in a negative value following adaptation. The net modulatory effect of $x_0$ on the $x_1 \rightarrow y$ pathway inverts the sign of gradients when $x_0$ is positive. The net gradient of $w_{hy}$, previously 0, goes net positive as a result of the modulation. (e) Final stable state with correct response in $y$.

multiplication operation and the nonlinear $\sigma_1$. The two terms in modulatory nodes will have distinct delta values, $\delta_{x,i}$; and they will form in-edges distinctly for these two terms. We assume that at the time of ENC, the original source $i$ connects to term 0 of the new node $k$ and no node is connected to term 1. We further set the bias of the first term to a fixed 0, and we set the weights of new edges as $w_{ik} = 1$ and $w_{kj} = w_{ij}$.

It is shown in the appendix that this design of a modulatory node satisfies our neutrality criterion. It can also be shown that immediately after this conversion, the deltas for term 1 of the new node are equal to the gradients of the original edge, $\delta^m_{k,1} = \partial C/\partial w_{ij}$. By ENC, we effectively transferred the weight gradients of the original edge (which cannot be treated as a vector for adaptive purposes, but only in terms of their average effect) to the deltas of a node (which can be treated as a vector that can create an adaptive change even if their average is 0). If a proper source $l$ can be found for term 1 of node $k$ that yields a nonzero $\partial C/\partial w_{lk}$, then the net gradient of the new edge $(k, j)$ will start going nonzero as state of term 1 of node $k$ is adapted under the influence of the gradients which are proportional to that of gradients of the original edge, hence getting out of what would be a local optimum had we have a static network (Figures 1c and 1d).

In the appendix, we show that the chain of ENC operations will continue, resulting eventually in nonzero net gradients, and hence adaptation will proceed across the network, as long as the following condition does *not* hold:

$$Cov(\prod_{x \in A} a_x^m, \frac{\partial C^m}{\partial w_{ij}}) = 0, \ \forall A \in P(N) \tag{2}$$

where $N$ is the set of all available candidate sources (at the very least, covering all input nodes) and $P(N)$ its power set. In the most relaxed case, this condition states that adaptation will proceed as long as there is any nonzero correlation between the gradient vector of the edge that we are trying to get out of adaptive exhaustion and any of the potential multiplicative combinations of the input nodes of the network. This is a condition much more strict than simply having a mean $\frac{\partial C^m}{\partial w_{ij}}$ as 0, as static networks would have; and we intuitively suspect, but did not verify mathematically, that it may correspond to a global optimum. The theoretical nature of this condition should not be forgotten, however, limited by finite step sizes and practical considerations.

To limit the number of GPs executed and limit network complexity increase to when it would be absolutely necessary, we can introduce a *priority ordering* mechanism, which chooses a limited number of GPs among all that are possible at a particular instant, preferring some over the others. The choice of such a priority ordering scheme can be made in many ways depending on designer priorities. For our implementation, we prioritize low complexity and hence introduce a quite restrictive scheme; in which we perform a single generative process per step among the whole input pathway of an output node, if and only if all the components on this pathway have their immediate AP exhausted. Detailed description of this algorithm can be found in the Appendix.

We refer to the method of structural adaptation in this section as *DIRAD* (from *directed adaptation*).

## 4 NOVELTY DETECTION VIA PREDICTION VALIDATIONS

Our aim now is to create a network that can detect whether the samples observed currently by the network are actually among the same type that it had adapted to or if they are new, causing unexpected responses by the network. For that, we present the *PREVAL* (from *prediction validation*) method that can predict the states of the nodes in the network using information from higher levels of computation, and then uses mismatches in these predictions to detect novel encounters, and finally use this information to realize continual adaptation in a system with multiple models.[2] Below, we describe of components of PREVAL.

**L0 and L1 networks**  In a supervised learning task, let *L0 network* be our task network. Suppose that upon accomplishment of a designer-specified condition (e.g. errors no longer decreasing), L0 is *stabilized* - i.e. no more parameter updates or generative processes, the response of every node to a given input is fixed. In PREVAL framework, we create a new network, a directed acyclic graph (DAG) within itself, the *L1 network*, following the stabilization of L0. Target nodes of L1 are all the nodes (including inputs) in L0 except output nodes, and its task is to predict the states of these target nodes as computed by L0 in response to a given input. L1 can use as inputs any of the nodes in L0, potentially including output nodes, as long as the following condition is not violated: *For any node $n_0 \in L0$, no node $n_1 \in L0$ can have a path to $n_0$ via L1 if it also has a path to $n_0$ via L0* - making sure that only the nodes at a higher level of computation predict the states of those in the lower levels of computation, i.e. abstract information predicts the concrete observation, not vice-versa. Hence in L1, we prevent simply performing the trivial replication of the pathways in L0. Given this new description of inputs, outputs, targets, and the additional constraint regarding connectivity; L1 can be adapted as usual with DIRAD and stabilized in the same manner as L0.

**Multiple models**  We define a *model* as any system within our framework that has a particular response pattern to the input. In our implementation, we interpret models as distinct, unrelated networks; though alternative conceptualizations, such as a subset of connections or subnetworks expressed within one network, or networks that are duplicated and differentiated from one another, are also possible and everything in this section applies to them as well.[3] In our conceptualization, the system consists of a dynamic number of models. Below, we present a method using the outputs of L1 networks that is able to (i) detect new tasks that show deviation from the structure of previous data, on which existing models were adapted, and create new models for these new tasks; and (ii) in the observation of new data, can choose among the existing models the one that is best-matching for processing that data, without needing to observe the target outputs. Irrespective of the definition of a "model", when a system can create new models to process automatically-detected new data belonging to different tasks, one can said to have prevented DA in the system since there is no more overwriting or loss of information. If the system can, furthermore, assign newly-encountered data to proper model among the set of models (for different tasks) it has available, then we can say that it is able to retrieve this information which was protected, and hence (in sum) is capable of continual learning.

---

[2]PREVAL can possibly be interpreted within the predictive coding framework (Millidge et al. (2021); Spratling (2017)).

[3]We experimentally saw that while adaptation was faster when adapting via the addition of connections to a single network; the final performance was better with new networks per detected task.

**Validation of models** An adapted L1 network can provide us with the mismatch information between the actual state of an input or internal node and its predicted state based on the state of the network at higher levels of computation; which can in turn be used to *validate* whether a new sample (during deployment) or batch (during ongoing adaptation) is consistent with what is expected based on the data that the current model was adapted to. Notice that this pertains to the conversion of a continuous metric (amount of mismatch in L1 nodes) to a discrete one (model validated/invalidated on a sample). Furthermore, when one is comparing multiple models in this regard (finding the model that "best matches" to the sample), there is no one-to-one correspondence between different models since they will have distinct L1 networks, with different targets possibly differing even in the order of magnitude of their numbers. Hence, there is no single best way to perform this validation. Here, we describe (and use in our experiments) one particular framework, recursive processes of validation across the hierarchy starting from node responses to whole batches.

At time of stabilization, we classify a target node $n$ of the L1 network as *confidently predicted (CP)* if the mean prediction error (PE) of this node across the last batch is lower than a threshold $T_{CP}$; and record the mean $\mu_n$ and standard deviation $\sigma_n$ of observed PE that node across the last batch. When processing a new sample, the response of the model (and hence retrospective predictions of L1 target nodes) are obtained. We classify the CP node $n$ in a model as *conflicted* if the observed PE in that node for that sample is larger than $\mu_n + T_{conf}\sigma_n$ for a preset multiplier $T_{conf}$. A model is said to be *validated* on a sample if, in its response to that sample, the ratio of conflicted nodes to total number of CP nodes $N_{conf}/N_{CP} < T_{SV}$ for a preset threshold $T_{SV}$.

During stabilization of a model $M$, we record the number of invalidated samples in the last batch that the model observed, $R_{IS}^M$. During adaptation, model validation is performed over whole batches. A model is said to be *validated* on a batch if the total ratio of samples invalidated within the batch for this model do not exceed $(1 + \epsilon_{IS})R_{IS}^M$ where $\epsilon_{IS}$ is a small margin allowed on top of estimated ratio.

**Recall and new task detection** During adaptation, if we have a still-adapting (non-stabilized) model, we always process new batches with that model. If all models are stabilized, then we perform the validation of the batch across all models. If there is a model for which batch is validated, that model (or the one with least ratio of invalid samples if there are multiple) is chosen to process it. If there is no such model, we create a new model starting from L0 adaptation stage, corresponding to the detection of a new task.[4] During deployment/test, we process data on a per-sample basis. For each sample, all models are checked and the sample is processed by the model that is validated on it (or the one with least ratio of CP nodes if multiple there are multiple), without new model creation.

This process is simplified and illustrated on Figure 2. With PREVAL, the capability for continual adaptation (without storage and replay of past data) is decomposed into the capabilities of detection of new tasks and of creating new network components (either on top of existing, without affecting pre-existing pathways' functionalities; or from scratch) and being able to deduce by which of those components an observed instance should be processed. Although other solutions proposed against DA, including those that work on fixed topologies, can alleviate it to a degree in certain tasks; we think that any definitive solution that can solve the problem at its root should be operating within this decomposition made with PREVAL.

DIRAD makes possible the implementation of PREVAL framework, which wouldn't be practical to construct using fixed topology layered networks (FTLNs) like fully connected NNs. The reason for that is twofold: One, we want to predict not just input nodes, but also internal nodes - which is not possible with FTLNs except by creating separate multi-layered overparameterized networks for each layer of the original already-overparameterized network - highly unscalable. With DIRAD, we promote minimal network complexity by design in all networks. Two, architectural limitations: If we use information bottlenecks, e.g. autoencoders, we have compression that can limit prediction, which we do not want since our requirement is as accurate prediction as possible without concern about compression. With no bottleneck, however, there is a risk of decaying into triviality (e.g. each input determining its prediction directly via a network transformation that is equivalent to identity; plausibly among the probable solutions in high-dimensional, difficult prediction tasks). PREVAL

---

[4]We assume that a batch for one task is available to the system until the end of its adaptation, and no new task is provided until system is stabilized for current one. This can easily be realized if computation power and temporary storage cost of a batch are not limiting factors, which is seldom the case in today's systems.

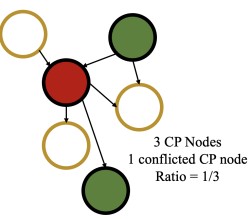

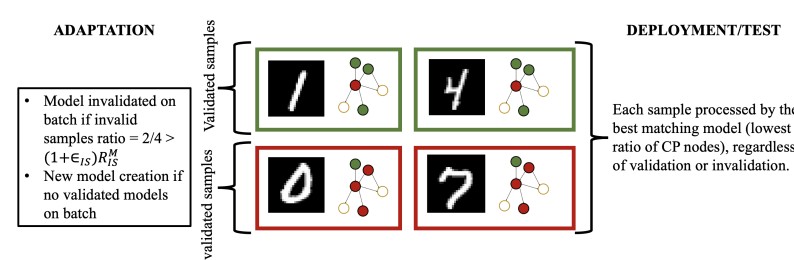

(a) For each sample, the responses of networks are obtained. Among the CP nodes (outlined in black), we take the ratio of those that are invalidated (red) as a measure of mismatch for the sample for a given model.

(b) During adaptation, if there is a standing adapting model, we process the batch by that model. If not (the case shown on figure), we process the batch by a model that is validated on a batch, which is the case if the ratio of invalidated samples (outlined red) to the total number of samples is greater than a threshold. If no model is validated, a new adapting model is created for the task. During deployment/test, each sample is processed by the best-matching model regardless of validation.

Figure 2: Simplified representation of PREVAL flow.

with DIRAD prevents unnecessary bottlenecks while giving the prediction of a node access to any info it can need without limitation, except for explicitly excluding those that would result in trivial predictions - this overcomes both limitations with FTLNs discussed. Furthermore, while the designer can choose the target dataset to yield an even distribution among different classes; this cannot be satisfied with the state of the internal nodes predicted back by L1 - the discerning among different states of a node may be reliant upon recognition of a small number of outliers for that particular node. An adaptation method based on statistical fine-tuning would have difficulty capturing these nuances as conditioned on the states of other variables in the network, which is not the case with DIRAD.

## 5 EXPERIMENTS AND RESULTS

**Setup**  In this section, we present experimental results with DIRAD and PREVAL. We use a digit classification problem with task change on MNIST and Fashion MNIST (F-MNIST)[5], where different tasks are defined as different classes (digits) enabled. For each experiment run, the process is as follows: Two classes are chosen randomly among the 10 classes in the dataset. We call this choice of a subset of classes from the whole dataset a *task*. The system adapts to the task at hand, first of its L0 network and then of its L1 network, until it reaches the stabilization of L1 network (detected via non-decreasing errors in all target nodes). After L1 stabilization, the task changes by choosing two new digits at random that were not among the previous tasks, and the new task is adapted to in the same manner, with potential model changes and additions as they are executed by the PREVAL flow. This process happens for 3 tasks (2 changes). We choose $T_{CP} = 0.05$, but provide additional results with different values on Appendix A.6. Detailed choice of other parameters is provided in Appendix A.5. We don't perform any ablation analysis (except the basic distinction of DIRAD and its augmentation with PREVAL) since all components of the algorithms are required for their basic operation.

**Metrics**  For DIRAD, we present the progression of mean squared error throughout training, first for L0 then for L1 networks. For PREVAL and continual learning baselines; at the end of each task, we test the *retrospective prediction performance* of the system on a test set formed of *all* the classes that belong to *all* the previous tasks that the system was exposed to. In other words; samples from only one task is shown to the network at a time during training, but samples from all the preceding tasks are tested on during testing.

---

[5]We downscale images to 14x14 pixels for computational purposes. Notice that this makes the classification task more difficult, not easier (and is partially responsible for a somewhat lower single-task accuracy compared to NNs); but limits number of L1 targets.

**Baselines**   We compare our method with three baselines: (1) Elastic Weight Consolidation (Kirkpatrick et al. (2017)) (EWC), (2) Progressive Neural Networks (Rusu et al. (2016)) coupled with Familiarity Autoencoders (Hadsell et al. (2020)) (PNN-FAE) for task assignment, (3) Memory Aware Synapses (MAS) for continual learning (Aljundi et al. (2019) - we used the variant in paper without replay buffer as is assumed in our problem setup, which cannot operate without task boundaries). Details and parameters for these implementation are in Appendix A.5. Note that while these methods were developed for information retention, neither is capable of detecting new tasks autonomously. Instead, they rely on external signals to indicate when a task change occurs. In contrast, PREVAL is designed to independently recognize and adapt to new tasks without external specification. As a result, PREVAL's capabilities are inherently superior, and in performance comparisons, these methods should be seen as upper limits. They provide an idea of what can be achieved by existing methods that lack the ability to recognize task changes or detect new tasks. As mentioned earlier, no comparable baseline in the literature that we know of demonstrates recall, task change recognition, and new task detection without external intervention or the use of stored past samples (i.e., replay).

**A note on higher dimensional tasks**   MNIST and Fashion MNIST are standard benchmarks for ML methods, though it is known that current NNs can handle more complex datasets in non-continual settings. Our methods, however, are more complex to implement, and experiment runtimes increased significantly with higher-dimensional tasks. As such, these results should not be seen as a direct comparison of DIRAD/PREVAL to the full non-continual capabilities of NNs. Instead, they demonstrate the core operational features (which are qualitatively more comprehensive than those available in existing methods, due to the possibility of recall and new task detection without external input) on recognized but simple benchmarks. The discussion on computational complexity and future directions is in the next section.

### 5.1   Results[6]

**Single-task adaptive performance**   Figure 3 shows a typical progress of adaptation on a single task, over course of both L0 and L1 adaptations, for MNIST. Across trials, it is consistently seen that errors for output nodes decrease to near-zero values during L0 adaptation; and to a non-zero value of $\approx 0.05$ per node during L1 adaptation - showing that, as expected, not every L1 target can be predicted reliably and a distinction of those that can be predicted confidently is needed.

Figure 3 also demonstrates the progression of network complexity for a single task. L0 adaptation can solve the task, in this run, with only 6 (hidden) nodes and 15 edges. This order of network size (¡20 nodes and ¡50 edges; and usually towards the lower end of this range) is observed in all our experiments with 2-class classification. This corresponds to a number of edges at least two orders of magnitude lower than those of a fully connected NN - a NN with a single hidden layer of 16 neurons (an optimistic minimum size estimate for a task like MNIST) would need 3296 edges, more than 100 times the typical network complexity of DIRAD. The ratio of complexities between an NN and a solution found by DIRAD can be expected to hold for tasks of any complexity, due to DIRAD's guarantees of finding a good solution with minimum size and NNs' reliance on overparameterization, as discussed in the preceding sections. However, note that L1 adaptation requires a much greater complexity of the network, more than 10-fold increase in edges, fueled by both the increased number of target nodes ($\approx 200$ nodes vs. 10 output nodes of which only 2 can be active) and the fact that the targets themselves are more difficult to predict (each target can take arbitrary values in $(0, 1)$ vs. one-hot vectors in classification). This suggests that doing the task of L1 network with a NN (setting aside functional difficulties of defining this task with them in the first place) would require a vast number of parameters to have a shot in demonstrating a reasonable performance.

**Continual adaptation**   To demonstrate the continual adaptation performance, we provide the average net retrospective prediction performances (across 8 runs and across all classes) of the systems with different settings on Table 1 (values outside parentheses). Recall that these are test performances including the classes enabled in both the current and earlier tasks. To provide a lower bound on performance; in task index $X$ one would expect an accuracy of $1/2X$ for completely random classification, and of $1/X$ in a network which can accomplish the latest task perfectly but has "for-

---

[6]Full results, decomposed by individual class accuracies in individual runs, can be found in the Appendix.

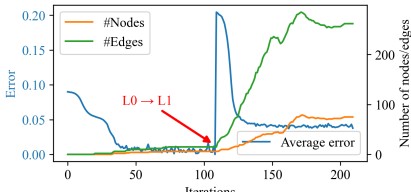

Figure 3: Sample progress of adaptation and network size on a single task on MNIST (2-class classification, 6 & 7). The average (mean squared) error is across all target nodes, i.e. output nodes during L0 adaptation and become (transition marked) the L1 target nodes during L1 adaptation.

gotten" (as is typically the case without measures against DA) the others. On the upper bound, we have a case in which all the tasks would be discerned perfectly, yielding approximately the accuracy it yields after only Task 1. We see that with PREVAL, the first task can be performed with a high accuracy of $\approx 96\%$ for both MNIST and F-MNIST, while each additional task causes a reduction in accuracy. Still, respectively for MNIST and F-MNIST, we see an accuracy of 79% & 84% after second task and 73% after third task, considerably higher than random (25%-17% resp.) and what one would expect for a perfect response only for the latest (50%-33% resp.). For MNIST, PREVAL is able to detect all task changes, while for F-MNIST, one change was missed in one run. There is no large performance difference between MNIST and Fashion MNIST for PREVAL on average, suggesting that datasets of varying complexities can be handled both.

In comparison with baselines, we see that EWC fails in continual learning as provided by incremental classes as the previously-learned task is eventually completely forgotten with the new task. PNN-FAE can retain the same performance across all tasks for MNIST, and to some degree in Fashion MNIST, close to PREVAL's performance[7]. MAS without replay buffer succeeds in retention for MNIST but fails in most trials in Fashion MNIST, giving a net accuracy lower than that of PREVAL. In short; is seen that PREVAL can operate on par with or surpassing existing methods (in all cases except PNN+FAE and MAS on MNIST, which show near-perfect retention), on top of the qualitatively superior capability of being able to detect task changes and recognize new tasks (which is not done by baselines).

Table 1: Average test accuracies (8 runs) for each $T_{CP}$ value. Entries under TX represent the network's net accuracy in classifying all classes that belonged to the tasks including and before index X (e.g. T2 is the accuracy on four classes, two from task 1 and 2 each); hence values under T1 can be also interpreted as single-task performance. Values inside parentheses are standard deviations. Number of runs with non-detected tasks is provided under entry "ND" (only for PREVAL), also included in the averages. Inside the parentheses are standard deviations. Recall that PREVAL's performance includes both task change detection and new task recognition, whereas these aspects are externally provided to the baseline methods.

| Task | $T_{CP}$ | T1 | T2 | T3 | ND |
|---|---|---|---|---|---|
| MNIST | PREVAL | 0.96 (0.026) | 0.79 (0.056) | 0.73 (0.061) | 0/8 |
| | Baseline: EWC | 1.00 (0.001) | 0.50 (0.010) | 0.33 (0.041) | NA |
| | Baseline: PNN+FAE | 0.98 (0.005) | 0.98 (0.003) | 0.98 (0.006) | NA |
| | Baseline: MAS | 0.99 (0.01) | 0.99 (0.01) | 0.99 (0.01) | NA |
| Fashion MNIST | PREVAL | 0.96 (0.058) | 0.84 (0.055) | 0.73 (0.077) | 1/8 |
| | Baseline: EWC | 0.96 (0.061) | 0.50 (0.016) | 0.34 (0.061) | NA |
| | Baseline: PNN+FAE | 0.90 (0.054) | 0.82 (0.098) | 0.79 (0.097) | NA |
| | Baseline: MAS | 0.87 (0.21) | 0.56 (0.38) | 0.49 (0.42) | NA |

---

[7]Note also that the autoencoders used to obtain PNN-FAE results are very complex compared to the L1 networks of PREVAL, with 7 layers and more than 40,000 parameters.

## 6 DISCUSSION AND CONCLUSIONS

**DIRAD**   A notable limitation of DIRAD lies in its computational complexity, which is not entirely mitigated by the architectural simplicity of the resulting networks. As previously mentioned, datasets with higher dimensions than MNIST and Fashion MNIST exhibited a marked increase in computation time. This heightened processing time is attributed to the current implementation of DIRAD, which handles all network components and samples sequentially, even though both aspects are well-suited to parallel processing. To scale DIRAD to higher-dimensional tasks—similar to the advancements made in neural networks since 2010—a more efficient, parallelized approach is necessary. However, such an implementation is beyond the scope of this paper and outside the expertise of the authors. It is likely that a method like DIRAD/PREVAL will always remain somewhat more computationally intensive than fixed neural networks. We see this as a trade-off for the distinct advantages these methods offer, such as unsupervised continual learning and the potential reduction in future learning processes enabled by systems with such capabilities.

Moreover, network growth methods do not yet fully benefit from hardware acceleration techniques (such as GPUs) that are effective for fixed, overparameterized NNs. We see this not as an inherent limitation but as a temporary and necessary challenge. Fixed and static network architectures have fundamental constraints, as detailed in the paper, and future "AI hardware" will need to adapt to support growing or dynamic networks. We view methods like DIRAD as innovations that will shape the development of future hardware. In cases like this, software should not be restricted by current hardware capabilities but should instead drive advancements in hardware, especially in evolving fields like AI.

**PREVAL**   To the best of our knowledge, PREVAL is the first framework that can handle continual adaptation with high accuracy & retention of past information, while doing both new task detection & discernment among past tasks within a unified framework that does not require task labels anywhere in the flow. The degree to which this happens with PREVAL, however, is dependent upon the discernability of different tasks within the system; and is currently still lower compared to what would be in case of a perfect discernability. Further analysis and improvements (including those pertaining to computational efficiency, as we had to train PREVAL with high learning rates for reasonable experimental duration, which may have caused destabilizations) are required to identify how the remaining performance gaps can be closed as well.

We saw that using familiarity autoencoders for assignment of a newcoming sample to its task can result in better task discernment than PREVAL (specifically in MNIST in our experiments) together with another growing architecture like Progressive NNs. It may be possible to use a hybrid system, in which PREVAL is be used mainly for new task detection (which it is shown to be able to do reliably) and an autoencoder is used for assignment of samples to known tasks may use the best of both worlds and result in higher net performance of the combined system.

We used PREVAL with an interface in which different models were represented with different networks. This prevents destructive adaptation (DA), but does not harness the potential of transfer learning (Zhuang et al. (2020)) since networks are all created from scratch. As discussed; in case of need, the definition of a "model" can be modified to allow for this. Alternatives include adding capacity to a shared network while selectively stabilizing the previous pathways so that past responses won't be affected (possible without loss of expressivity potential with DIRAD), modifying existing components while storing alternatives in different models, and many others. There already exist methods that work via the addition of capacity to alleviate DA in continual learning scenarios (Rusu et al. (2016); Yoon et al. (2017); Terekhov et al. (2015)) - approaches like that can be applied to PREVAL without changes in basic system conceptualization in either side. PREVAL can also be used with methods are reliant on storing past structure (like (Kirkpatrick et al. (2017))), to give them a means of detecting new tasks. PREVAL has been designed to operate as a mechanism at a level above the network adaptation process. Hence, it can work in tandem with any method modifying network adaptation dynamics that also aim to reduce the effect of DA, or is geared towards any other purpose.

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

# A  APPENDIX

## A.1  FULL THEORETICAL DESCRIPTION OF DIRAD

This section presents the full theoretical description and development of DIRAD. Note that the related part in the main text is a shortened version of this discussion, stripped to its main points.

### A.1.1  STARTING POINT FOR DEVELOPMENT

We start with the assumption of a network with the specified input and output nodes, as they are defined by the task, and no *hidden nodes*. From that, we want to develop a network that (1) can complexify as needed, but no more than needed, to solve the task at hand; and (2) is not limited by statistical conflicts between different samples in a given batch in doing so. Note that the process described in this section is based on mechanisms that operate locally in individual component basis, and hence their operation does not depend on the existing network complexity (even if their outcomes do).

We call the processes that grow the network by introducing new components (nodes or edges) as *generative processes*. In designing our generative processes, we prioritized the principle of *neutrality*[8] of structural modification: For all existing nodes in the network, neither their responses nor the adaptive signals (gradients/deltas) they receive should change by a generative process in the network - all changes should occur under the influence of gradients, to make sure that they are not detrimental to the performance. This principle corresponds to different criteria for different generative processes. Below, we describe the two main generative processes in our design: Edge generation and edge-to-node conversion.

### A.1.2  EDGE GENERATION

The first mechanism that we have to account for in our system is the edge generation mechanism. An edge $(i, j)$ with weight $w_{ij}$ from node $i$ to $j$ is a potential point of modification for a change in the activation value of its target node, affecting its state by multiplying the weight of its input:

$$a_j = \sigma(z_j) = \sigma\left( \sum_{i \in src(j)} w_{ij} a_i + b_j \right) \tag{3}$$

where $a_j$ is the state of node $j$, $z_j$ is the activation, sigma is a nonlinear function, and $b_j$ is the bias. Recognize that the contribution of a single edge to the activation of node $j$ is $+w_{ij}a_i$. For the principle of neutrality to hold, we want to make sure that a new edge $(i, j)$ does not change the value of $z_j$ before and after this operation, and hence during edge generation, we always initialize the weight of the edge as 0.

Recall that in NNs adapted with standard gradient descent, we have the gradient of a weight with respect to the cost associated with sample $m$ from a batch as follows:

$$\frac{\partial C^m}{\partial w_{ij}} = a_i^m \delta_j^m \tag{4}$$

$\delta_j^m = \frac{\partial C^m}{\partial z_j^m}$ are the gradients of cost terms with respect to activation $z_j^m$ of node $j$, a value that can be computed via applying multivariable chain rule starting from the output nodes and traversing the network backwards until the input nodes, forming what is called the backpropagation algorithm. Notice that the process for an NN with a layered & fully-connected (or any other mainstream fixed) topology is the same as the process for an arbitrary-topology network[9], like the one we assume

---

[8]Neutrality of individual changes, which open up the potential of an genome for further adaptive changes, are believed to be a core point in the evolvability of organisms. Interested readers are referred to Wagner (2011) for a organized overview.

[9]Given it is a directed acyclic graph (DAG), a condition that we assume in our description and enforce on generative processes in our implementation.

here: The $\delta$ values for a node can be computed only if the $\delta$ values for all its successors have been computed already, which gives us the needed order of traversal.

The gradient of a weight is then used to update the current edge weight:

$$w_{ij}[t+1] = w_{ij}[t] - \gamma \frac{\partial C}{\partial w_{ij}} \tag{5}$$

where $\gamma$ determines the magnitude/rate of adaptation per step, and total gradient for a given sample set of size $N_m$ is given by

$$\frac{\partial C}{\partial w_{ij}} = \frac{1}{N_m} \sum_{m \in M} \left( \frac{\partial C^m}{\partial w_{ij}} \right) = \frac{1}{N_m} \sum_{m \in M} (a_i^m \delta_j^m) \tag{6}$$

The magnitude of the value $\frac{\partial C}{\partial w_{ij}}$, which can be computed with the knowledge of source states and target deltas even without the existence of an edge between them, can be interpreted as the immediate *adaptive potential* (AP) of a given edge: It represents how much cost is expected to decrease after a single update step of that edge.

We can condition the formation of an in-edge to a target node upon a quantified need for it. Verbally speaking, an edge would be required when a node's activation requires a change for proper adaptation in different samples, yet the net pressures from the aggregate samples do not result in a net gradient in either the node's bias or the in-edges of the node.

More precisely, we define the *immediate AP* of an edge as the net gradient that this edge's weight gets over a given batch, i.e. $\frac{\partial C}{\partial w_{ij}}$ as defined above. We say that the immediate AP of an edge $(i,j)$ is *exhausted* if $\frac{\partial C}{\partial w_{ij}} \approx 0$. Analogously, we say that the immediate AP of a node $j$ is exhausted if $\frac{\partial C}{\partial b_j} = \frac{\partial C}{\partial z_j} = \delta_j \approx 0$ and the immediate AP of all its in-edges is exhausted as well. Furthermore, we can define the *total AP* of a node as the sum of the magnitudes of its delta terms (the adaptive pressures for each sample), $\sum_m |\delta_j^m|$, which gives a measure of the total adaptive gain that can be obtained by a change in the activation of that node $z_j$, if it can be exploited. With these definitions, we generate an edge for a node $j$ if the two following conditions hold at the same time:

1. The immediate AP of $j$ is exhausted.
2. The total AP of $j$ is not exhausted.

Notice that this edge generation condition ensures that a new edge will be generated (i.e. the network will complexify) only when the required change cannot be realized with adaptation of the existing components, preventing unnecessary complexification.

After defining the formation criterion, we need a means to choose the source of the edge. For that, Eq. 6 gives us a means to quantify the AP of an edge. Following that, when we generate an in-edge for a node $j$, we choose the source of the edge to be the one that maximizes the value $|\sum_m (a_i^m \delta_j^m)|$ among the candidates (which are, in our implementation, all nodes that do not create cycles; but it can equally well be chosen to be a subset of nodes). One way to interpret this is that we are trying to maximize the magnitude of the vector $\mathbf{a_i} \cdot \delta_{\mathbf{j}}$ where $\mathbf{a_i} = (a_i^0, a_i^1, ...)$ and $\delta_{\mathbf{i}} = (\delta_i^0, \delta_i^1, ...)$ are vectors of states and deltas of the corresponding nodes over the whole batch. Hence, when an edge is being formed, it can be thought of being formed with source as the node whose states/responses are *best aligned* with the change required by the target of the connection.

### A.1.3 Edge-node conversion and resolving statistical conflicts

Adaptive potential exhaustion is not a problem for nodes only, but for edges as well. The edge-to-node conversion (ENC) mechanism we develop addresses this problem, while also giving us a means of generating the second class of components for our network, i.e. the (hidden) nodes.

Analogous to the total AP of a node as defined in the previous section, we define the *total AP of an edge (i,j)* as the sum of magnitudes of gradients for this edge's weight: $\sum_m |\frac{\partial C^m}{\partial w_{ij}}|$. Again, this quantifies the total adaptive gain that can be obtained from a given edge if its weight could

be properly modified for each of the samples in the batch. With a classical edge such as those in neural networks, for example, this is not possible: Optimization of neural networks is fundamentally based on finding a statistical trade-off between the samples and stabilizing the weights of the edges in a point where the net gradient $\frac{\partial C}{\partial w_{ij}} \approx 0$ (i.e. the immediate AP of edge is exhausted) despite potentially high total AP. ENC mechanism, described in this section, specifically addresses these situations. More precisely, we perform the ENC operation on edge $(i, j)$ when the following two conditions hold:

1. The immediate AP of $(i, j)$ is exhausted.

2. The total AP of $(i, j)$ is not exhausted.

Like edge generation conditions, the ENC conditions make sure that we complexify only when the adaptation of existing components are insufficient to exploit the adaptive potential in the network and current samples.

When these conditions are satisfied, we convert the original edge $(i, j)$ into a path formed of a new node $k$ and edges $(i, k)$ and $(k, j)$ - in other words, we introduce a *point of modulation* to where the edge previously stood. Our goal with this point of modulation is to modulate the opposing yet nonzero gradients that the original edge was once under the influence of, in a manner that aligns them so that their adaptive potential, as quantified by their total magnitude, can be fully utilized without falling for statistical stable points.

To be able to realize this modulation, we design our node $k$ (and hence, any node created via the ENC mechanism - therefore, any node that is neither input nor output in our method) slightly differently from what would be a node in a neural network. In particular, we call these nodes *modulatory*, formed of terms 0 and 1, with distinct biases, sources, activations and states; and whose states are multiplied to result in the final state. The functionality of any modulatory node $x$ as such is described by the following equations:

$$a_x^m = a_{x,0}^m a_{x,1}^m = \sigma_0(z_{x,0}^m)\sigma_1(z_{x,1}^m) \tag{7}$$

$$z_{x,i}^m = \sum_{y \in src_i(y)} w_{yx}a_y + b_{x,i} \tag{8}$$

All the terms here are analogous to those in Eq. 3, but those with subscripts $x, i$ instead of $x$ refer to *node x, term i*, instead of just *node x*. We also assume two transfer functions $\sigma_0$ and $\sigma_1$. Likewise, the two terms will have distinct delta values, $\delta_{x,i}$; and modulatory nodes will form edges separately for these two terms based on their corresponding term deltas.

We assume that at the time of ENC, the original source $i$ connects to term 0 of the new *converted node k*, and at first no node is connected to term 1 of *k*. As a simplifying design choice, we also let $\sigma_0$ to be linear for modulatory nodes, $\sigma_0(z) = z$. Hence, at the time of conversion, the actual value of the source ($i$ in our example) of a modulatory node ($k$ in our example) can be reliably transmitted to the state of the new converted node, without losing the nonlinearity known to be required in complex adaptive networks since we have both the multiplication operation and also $\sigma_1$, which we leave as nonlinear.

The rest of the properties of a modulatory node will be defined by the condition of neutrality. (In what follows, for better clarity, we omit from $a$, $z$ and $\delta$s the superscript $m$ denoting a specific sample in a batch. All of the discussion below is concerned with the states of the nodes for a single particular sample. We also consider a single activation term in target node, but all the derivations remain the same if we assume that we are concerned with activation of term 0 or 1 of the target instead.) Before ENC, the contribution of node $i$ to the state of node $j$ is:

$$z_j^i = w_{ij}a_i \tag{9}$$

After ENC, it is mediated by $k$ (which initially has no connection other than node $i$ connecting to its term 0):

$$z_j^i = w_{kj}a_k = w_{kj}a_{k,0}a_{k,1} = w_{kj}z_{k,0}\sigma_1(z_{k,1})$$
$$= w_{kj}w_{ik}a_i\sigma_1(b_{k,1}) \tag{10}$$

For neutrality, we want these two values to be the same. Therefore,

$$w_{ij} = w_{kj}w_{ik}\sigma_1(b_{k,1}) \tag{11}$$

Among the many ways to realize this condition, we choose to let $\sigma_1(b_{k,1}) = 1$ and $w_{ij} = w_{kj}w_{ik}$. For the latter, we let the weight of the in-edge of ENC be $w_{ik} = 1$ and of the out-edge be $w_{kj} = w_{ij}$, the original weight[10]. For the former, we need to choose a suitable transfer function $\sigma_1$ and a suitable initial bias $b_{k,1}$ that makes $\sigma_1(b_{k,1}) = 1$. Again, among infinite alternatives, we decide to set $b_{k,1} = 0$ and $\sigma_1$ to be a scaled and shifted logistic function:

$$\sigma_1(x) = 4\frac{1}{1 + e^{-Kx}} - 1 \tag{12}$$

Besides yielding $\sigma_1(0) = 1$ and hence realizing our neutrality condition, this function has the desirable property of taking values in the range $(-1, 3)$ - being able to take negative values means that in multiplication, it is able to invert the sign of the previously-opposing gradients. (In what follows, we reintroduce the superscript $m$ denoting individual samples from a batch.)

Note that before conversion, the edge had gradients $\frac{\partial C^m}{\partial w_{ij}} = a_i^m\delta_j^m$. After conversion, the out-edge, which was initialized to the weight of the original edge, will have gradients $\frac{\partial C^m}{\partial w_{kj}} = a_k^m\delta_j^m = \sigma_1(z_{k,1}^m)a_i^m\delta_j^m$ - i.e. the original gradients multiplied by the modulatory term. (Similar derivation will show that $\frac{\partial C^m}{\partial w_{ik}} = w_{ij}\sigma_1(z_{k,1}^m)a_i^m\delta_j^m$ - the same value multiplied with the original weight. We keep our focus on the out-edge in the narrative, but all apply to the in-edge as well.) Hence, provided that we can adapt the response of term 1 of node $k$ properly, we can cancel or even invert the opposing terms that once made $dC/dw_{ij} \approx 0$ and hence get a net positive or negative $\frac{\partial C}{\partial w_{kj}}$. This is to be done by generating edges for term 1 of node $k$ that can selectively modulate the node to be positive or negative depending on the sign of the gradients. To see if this will be the case, we check the adaptive signals that term 1 of $k$ receives. Applying chain rule to Eq. 8 and substituting the values we chose will show us that, at time of ENC:

$$\delta_{k,1}^m = \frac{\partial C^m}{z_{k,1}^m} = \sigma_1'(z_{k,1}^m)w_{ij}a_i^m\delta_j^m = Kw_{ij}a_i^m\delta_j^m \tag{13}$$

In the last step, we replaced $z_{k,1}^m = 0$ (its initial value), which can be shown to yield $\sigma_1'(0) = K$. The initial deltas for term 1 are indeed proportional to the original weight gradients. Hence, we have effectively transferred the weight gradients of the original edge (which cannot be treated as a vector for adaptive purposes, but only in terms of their net/average effect) to the deltas of a node (which can be treated as a vector that can create an adaptive change even if their average is 0, as discussed in the previous section). We, furthermore, choose our free parameter $K = 1/w_{ij}$ for each newly-created node to make sure that $\delta_{k,1}^m = \frac{\partial C}{\partial w_{ij}}$ under all circumstances, to avoid issues of diminished deltas with high gradients for low original weight magnitudes. If a proper source $l$ can be found for term 1 of node $k$ (by the edge generation mechanisms in the preceding section) that yields a nonzero $\frac{\partial C}{\partial w_{lk}}$, then the net gradient of the new edge $(k, j)$ will start going nonzero as the state of term 1 of node $k$ is adapted under the influence of the gradients which are proportional to that of gradients of the original edge, hence getting out of what would be a local optimum had we have a static network. Note that this perfect correspondence between original gradients and the deltas of the new node only hold for immediately after ENC - as adaptation progresses, both the gradients of the new edge as well as

---

[10]In theory, any pair of $k_w w_{ij}$ and $1/k_w$ can be chosen here. In practice, however, we saw that choosing a $k_w$ different from 1 results in an uncontrollably decaying or exploding weights in the presence of multiple consecutive ENC operations with little adaptive change via gradients in between. Choosing $k_w = 1$ as we do better stabilizes observed weight values in the network.

$\delta_{k,1}^m$s will take new values, the latter because $\sigma_1'(z)$ will get different, potentially sample-dependent values. However, this breakdown of alignment will happen gradually as the network components adapt, and the net gradients for the edges will initially increase in magnitude and get out of their exhaustion at the first steps (and later re-exhaustions will simply trigger new ENC processes, as this process proceeds locally where needed over the course of network adaptation).

What happens if we cannot find a source $l$ for $k$ term 1 that yields a nonzero $\frac{\partial C}{\partial w_{lk}}$, anywhere in the network? Edge generation process, by our design, will nevertheless form the edge $(l, k)$ but with a zero-gradient (exhausted immediate AP). If, however, the total adaptive potential of new edge $(l, k)$ is nonzero, then this new edge itself will undergo the ENC operation described here, creating some other node $k_1$ and edges $(l, k_1)$ and $(k_1, k)$. We will then look for a source $l_1$ for term 1 of $k_1$ in a similar manner, and with the criteria that we defined in the beginning of this section, this process will go on recursively, until we find *some* source at an arbitrary depth of recursion that yields a nonzero net gradient for its edge - whose adaptation will start a chain process that will start aligning the gradients for all the edges formed before it as a part of this recursive ENC chain, getting their immediate APs out of exhaustion and proceeding with adaptation.

In fact, we can semi-formally define a minimum condition that guarantees that adaptation will eventually proceed under influence of nonzero adaptive gradients after a chain of ENC operations. Notice that if an edge $(l, k)$ has a zero net gradient, we have:

$$
\begin{aligned}
\frac{\partial C}{\partial w_{lk}} &= \frac{1}{N_m} \sum_m \frac{\partial C^m}{\partial w_{lk}} = \frac{1}{N_m} \sum_m a_l^m \delta_{k,1}^m \\
&= \frac{1}{N_m} \sum_m a_l^m \frac{\partial C^m}{\partial w_{ij}} = 0
\end{aligned}
\tag{14}
$$

Since $\frac{\partial C}{\partial w_{ij}} = \frac{1}{N_m} \sum_m \frac{\partial C^m}{\partial w_{ij}} = 0$ (condition for ENC), it will also hold that $M \sum_m \frac{\partial C^m}{\partial w_{ij}} = 0$ for any scalar multiplier $M$. Hence:

$$
\sum_m a_l^m \frac{\partial C^m}{\partial w_{ij}} = 0 = \sum_m M \frac{\partial C^m}{\partial w_{ij}}
\tag{15}
$$

$$
\sum_m (a_l^m - M) \frac{\partial C^m}{\partial w_{ij}} = 0
\tag{16}
$$

Since this must hold for all $M$, it must also hold if we choose $M$ to be the expected value of $a_l^m$, $\bar{a}_l$.

$$
\sum_m (a_l^m - \bar{a}_l) (\frac{\partial C^m}{\partial w_{ij}} - 0) = 0
\tag{17}
$$

Since the mean of $\frac{\partial C^m}{\partial w_{ij}}$ is 0, the condition for immediate exhaustion of the first formed edge $(l, k)$ is:

$$
Cov(a_l^m, \frac{\partial C^m}{\partial w_{ij}}) = 0
\tag{18}
$$

But this won't be just the first formed edge. For a subsequent ENC operation on $(l, k)$, with generated node $k_1$ and some other edge from another node $l_1$, we'll analogously have:

$$
\frac{\partial C}{\partial w_{l_1 k_1}} = \sum_m a_{l_1}^m \frac{\partial C^m}{\partial w_{lk}} = \sum_m a_{l_1}^m a_l^m \frac{\partial C^m}{\partial w_{ij}} = 0
\tag{19}
$$

Similarly, this condition will require for immediate AP exhaustion:

$$Cov(a_{l_1}^m a_l^m, \frac{\partial C^m}{\partial w_{ij}}) = 0 \tag{20}$$

We can extend this process to the multiplication of all the nodes $(l_2, l_3, ...)$ that we'll take as source in this chain of ENC, which will be a subset of all nodes in our network that are available as source candidates. Hence, if we let $N$ be the set of all available candidate sources (which, at the very least, always covers all input nodes available), and if $P(N)$ is its power set, then the chains of ENC operations will result in a nonzero net gradient (and then getting the other gradients they modulate out of their zero-values and kickstarting the adaptive process again) as long as the following condition does *not* hold:

$$Cov(\prod_{x \in A} a_x^m, \frac{\partial C^m}{\partial w_{ij}}) = 0, \ \forall A \in P(N) \tag{21}$$

In the most relaxed case ($N$ only includes the input nodes), this condition states that adaptation will proceed as long as there is any nonzero correlation between the gradient vector of the edge that we are trying to get out of adaptive exhaustion and any of the potential multiplicative combinations of the input nodes of the network. This is a condition much more strict than simply having a mean $\frac{\partial C^m}{\partial w_{ij}}$ as 0, as static networks would have. We did not check whether this condition would guarantee a global optimum theoretically, and it is an open question for the researchers who would like to see if this is the case. It should be noted, however, that this condition only holds in the context of the theoretical analysis of this section - in practice, other factors such as acceptable complexity and error limits or discrete gradient updates can result in not being able to actually realize this potential within reasonable limits, if the problem is sufficiently difficult for that.

### A.2 SUPPLEMENTARY MECHANISMS AND PRACTICAL MODIFICATIONS FOR DIRAD

In this section, we describe some practical design choices we made for the implementation of the mechanisms of DIRAD described in the previous section.

#### A.2.1 DESTRUCTIVE PROCESSES

In practice, due to the strictness of removal conditions and the extremely targeted nature of generative processes, we almost never see a component removed from the network in DIRAD. We nonetheless describe our destructive processes here for completeness purposes.

We remove an edge if its weight is 0 and we remove a node with a probability of $0.3$ if it has no remaining out-edges.[11] Edges are protected against removal for 5 steps after their first creation. The probabilistic nature of node removal is to prevent the immediate destruction of complete pathways of interconnected nodes via recursive removal of them from their end-points, and instead to give the nodes within this pathway a chance to be able to participate in the responses of other nodes by acting as sources of in-edges generated by them. These two destructive processes of DIRAD are both neutral.

We have an additional destructive process specific to PREVAL and only during L1 adaptation: At time of stabilization of an L1 target node, we remove the predictive pathway of the node if it is not confidently predicted. This is simply because prediction of non-CP nodes have no further use for the system, and we remove them to cap the complexity increase, hence make it more scalable. The fluctuations in complexity on Figure 3 are due to this mechanism and not to DIRAD's destructive processes.

#### A.2.2 ACCEPTABLE MISMATCH

DIRAD mechanisms as defined above will keep complexifying as long as there is nonzero mismatch in the network. To avoid overcomplexification after obtaining reasonable performance, it is desirable

---

[11]Notice that in our implementation, we do not have parameter decay (i.e. L2 regularization) which would drive edge weights to 0 and probably result in more frequent removals.

to define a lower cut-off limit, below which any mismatch in the output nodes will be regarded as 0. For this purpose, we chose to regard an mismatch in magnitude at the outputs below 0.01 (in a possible scale of 0 to 1 - i.e. 99%) in an individual sample as 0.

$$|a_y^m - \hat{y}| < 0.01 \rightarrow \frac{\partial C^m}{da_y^m} = 0 \tag{22}$$

### A.2.3 GRADIENTS

Since the gradients and deltas of a real, discrete-update network will never be exactly 0, we need criteria for lower-limiting them and treating as 0. Notice that this is only required for generative processes to be able to occur (i.e. when checking exhaustion conditions), and does not need to be applied to other uses of gradients or deltas in the system.

For that, we use the following measures:

1. We define a $\delta_{min}$. The deltas below this value will be considered as 0 when checking exhaustion conditions and computing AP values. In our implementation, we choose $\delta_{min} = 0.01$.

2. We condition the immediate AP exhaustion of an edge on the ratio of total net gradient magnitude and the mean magnitude of the gradients in the two individual directions. Let $M^+$ be the set $\{m : \frac{\partial C}{\partial w_{ij}} > 0\}$ and vice versa for $M^-$. Then, ENC will occur if, for a chosen ratio parameter $R_1$:

$$\exists M \in \{M^+, M^-\} : R_1 \left| \frac{\partial C}{\partial w_{ij}} \right| < \left| \frac{1}{N_m} \sum_{m \in M} \frac{\partial C^m}{\partial w_{ij}} \right| \tag{23}$$

We chose $R_1 = 5$ in our implementation.

3. Furthermore, to speed up adaptation, we found it useful to always regard an edge as having exhausted immediate AP if $\left| \frac{\partial C}{\partial w_{ij}} \right| < R_2 |w_{ij}|$ - i.e. if it falls below a ratio of the standing weight magnitude. We chose $R_2 = 0.1$.

The operation of DIRAD is robust to the particular choice of these parameters. The values they take mainly serve to control the speed of adaptation by modulating the degree of exhaustion required to initiate generative processes.

### A.2.4 FREE PARAMETER IN TERM 1 TRANSFER FUNCTION

Recall that we chose the free parameter $K$ of $\sigma_1(.)$ as $K = 1/w_{ij}$ following an ENC operation. Since this value is not defined if $w_{ij} = 0$, we do not allow ENC to the edges with weights 0. To facilitate the edges from getting out of zero-weight situations, we add a small perturbation to $\frac{\partial C^m}{\partial w_{ij}}$ terms of the edges with weights 0 when computing the final $\frac{\partial C}{\partial w_{ij}}$, where the perturbation is of the form of a random noise with mean 0 and standard deviation $0.05 \left| \frac{\partial C^m}{\partial w_{ij}} \right|$ for each $m$.

With this framework, it is expected that the $K$ values (which control the steepness of the function) will initially be very large for small-weight edges. Very large $K$ values will result in rapid saturation of the created node, since the sigmoidal function response will saturate for all samples upon even the slightest change, and hence propagate back no gradients. To avoid that, after we initialize an initial $K = 1/w_{ij}$ upon ENC, we decay that value each iteration with a decay rate of 0.1 towards a stable point of $K_{final} = 1$, i.e. $K(t+1) \leftarrow K(t) * 0.9 + 0.1$. With that, we can transfer the gradients of an edge to the deltas of the converted node at the time of ENC, while still preventing premature saturation of the node. Notice that this decay of $K$ values, however, has the drawback of not being totally neutral and potentially resulting in a slow non-adaptive change in the network states as long as it continues. In practice, we did not observe that this creates a major issue in long-term performance; but if it does, choice of slower decay rates should alleviate it.

---

**Algorithm 1** V1 Priority ordering algorithm. Internal functions are not written in detail to prevent overcrowding, their mechanisms are as explained in the main text. Among those; functions *SatisfiesENCCondition* and *SatisfiesEdgeGenCondition* check whether their arguments satisfy the ENC and edge generation conditions, respectively; and functions *PerformENC* and *PerformEdgeGen* perform ENC and edge generation operations respectively. *PathwayAPExhausted* checks if the pathway starting from the argument node is exhausted, and *TotalAP* returns the total AP of the argument.

---

**Parameter**: $N$ Set of all target nodes
**Function** $V1PO()$
1: $N \leftarrow \{n : n \in N \land PathwayAPExhausted(n)\}$
2: $N \leftarrow OrderByTotalAP(N)$
3: **for** $n \in N$ **do**
4:    $GPFor(n)$
5:    $N \leftarrow \{n : n \in N \land PathwayAPExhausted(n)\}$
6: **end for**
**Function**: $GPFor(n)$
1: **while** True **do**
2:    $e \leftarrow HighestAPInEdge(currNode)$
3:    **if** $TotalAP(e) > 0$ **then**
4:      **if** $TotalAP(source(e)) > 0$ **then**
5:         $currNode \leftarrow source(e)$
6:      **else**
7:         **if** $SatisfiesENCCondition(e)$ **then**
8:            $PerformENC(e)$
9:            $return$
10:        **end if**
11:      **end if**
12:    **else**
13:      **if** $SatisfiesEdgeGenCondition(currNode)$ **then**
14:         $PerformEdgeGen(currNode)$
15:         $return$
16:      **end if**
17:    **end if**
18: **end while**

---

### A.3 DETAILED DESCRIPTION OF PRIORITY-ORDERING ALGORITHM

The detailed description of our priority ordering algorithm (Algorithm 1) is as follows: We take all of the target nodes (the ultimate sources of adaptive pressures from mismatches with their targets - e.g. output nodes in a classification task) whose immediate AP is exhausted for the whole of their response pathway (i.e. all nodes that have a path to that node and all the edges in these paths), and order them by decreasing total AP. As long as their immediate AP is exhausted (since a given node elsewhere in the network may be participating in the response pathways for multiple target nodes), we perform one and only one generative process (either an edge generation or an ENC) within the pathways for each of the target nodes. For each node, we start from the target node as our current node, and search over all its edges in the order of decreasing total AP. At an edge, if we find one whose source has nonzero total AP, we take that source node as our current node and restart the search from there. If there is no such node, we check whether the edge satisfies ENC condition. If it does, we perform ENC. If it does not, we move onto the next edge. If no edge neither satisfies ENC nor has a source that has nonzero total AP, then we return to our current node and generate an edge for it if it satisfies edge generation conditions. If it doesn't satisfy edge generation conditions as well, we return for this node without performing a generative process (since none suitable has been found).

As mentioned in the main text, the presence of a priority ordering mechanism is not indispensable for the operation of DIRAD processes. It is likely that adaptation will proceed faster without these restrictions imposed on generative processes, at the cost of increased complexity. If, furthermore, a priority ordering over GPs is going to be used, the exact form it takes can change depending

on the requirements of the system - our framework is very restrictive since we prioritize minimal complexity (due to the fact that we will predict the states of generated nodes after stabilization), but in an application where complexity requirements are more relaxed, one can use a less restrictive scheme.

## A.4 DETAILED EXPERIMENTAL SETTINGS

We use a version of MNIST resized to $14 \times 14$ pixels instead of the original $28 \times 28$ (except for one set, see below) in order to speed up our experiments. Notice that this makes the classification more difficult, not easier; yet makes L0-prediction with PREVAL operate on a more reasonable scale.

Throughout our experiments, we use a learning rate $\gamma = 1^{12}$ for both DIRAD/PREVAL and the baselines. For DIRAD, we also incorporate a *refraction period* (a period, initiated after the execution of a generative process, during which no other GP can take place) of 5 steps for each node and edge, in order to limit the speed of complexity increase. We stabilize the L0 and L1 networks automatically upon observation of no total prediction error decrease for 50 steps. We perform adaptation with batches of size 100 (2 classes), changing every step. Our test batch sizes were 300 (since at most, we can have 6=3x2 classes).

For the PREVAL threshold parameters, we use $T_{conf} = 1.5$, $T_{SV} = 0.01$, and $\epsilon_{IS} = 0.2$. $T_{CP} = 0.5$ except for results in section A.6.

For EWC and PNN-FAEs, we used a learning rate of 1 (same as we use for DIRAD), a training batch size of 32. Lambda parameter for EWC loss was chosen as 50 and 200 samples were used to compute Fisher information. For EWC training continued for 20 epochs while for PNN-FAE, 50 epochs. For both methods, the task network consisted of one hidden layer with 32 neurons (for PNNs, this corresponds to the size of a single new column). For FAE, we used an autoencoder of 7 layers with 120-100-64-32-64-100-120 neurons respectively. We used Doric library dor for implementation of PNNs, and the authors' own implementation for MAS. For MAS, we used a learning rate of 0.1 since the method failed completely with learning rate 1.

*Computation resources:* All experiments were run on a 2.4GHz 8-Core Intel Core i9 processor with 32 GB 2667MHz DDR4 memory. No GPU was used. Giving an accurate estimate for computation time is impossible since experiments were run in parallel to unevenly-distributed independent workloads; though a single independent run of PREVAL on three tasks (including growth of L1 networks) can be estimated to take in the range of a few hours.

## A.5 DETAILED EXPERIMENT RESULTS

On Table 2 to 3, we provide the results with different CP thresholds decomposed by runs and individual class accuracies. The individual class accuracies show the ratio of *true positives* for that class in the corresponding run. We note that in most of the runs we executed, the discernability of a small number of tasks was observed to be the primary factor contributing to reduction of accuracies (e.g. Table 2 in Appendix, run 2, class 5) while the other classes kept being recognized by a high accuracy. This suggests a degree of difference between different classes with respect to their discerneability. We provide on Table 4 the accuracies for individual classes, to see if there are those that are more difficult to discern. Indeed, we see that some classes (like 0, 6) are easier to discern compared to the others (like 8 or 3). This may correspond to digits of similar structural elements, which could result in PREVAL validating a sample belonging to one of them on the model for the other.

In addition to the results displayed on tables, we experimented with two supplementary deviations from the above-mentioned settings: With full MNIST ($28 \times 28$ instead of $14 \times 14$) and a larger batch size (300 instead of 100). The results with both were similar to those with our current settings, the only notable difference was $\approx 5\%$ higher single-task accuracy (i.e. after only the first task, and without additional tasks presented) with full MNIST. We do not include these results in detail to prevent overcrowding.

---

[12]Except for our results comparing different $T_{CP}$ values, see A.6.

Table 2: Full results on MNIST. Entries are formatted as "class index: accuracy", and are separated by commas for different classes.

| Run | T1 | T2 | T3 |
|---|---|---|---|
| 1 | 4: 0.953, 8: 0.96 | 4: 0.947, 8: 0.92, 1: 0.613, 6: 0.76 | 4: 0.8, 8: 0.78, 1: 0.6, 6: 0.66, 3: 0.68, 2: 0.82 |
| 2 | 2: 0.953, 8: 0.927 | 2: 0.827, 8: 0.693, 6: 0.613, 7: 0.773 | 2: 0.66, 8: 0.66, 6: 0.62, 7: 0.74, 3: 0.5, 0: 0.72 |
| 3 | 0: 0.987, 4: 0.993 | 0: 0.96, 4: 0.613, 7: 0.813, 5: 0.76 | 0: 0.9, 4: 0.66, 7: 0.92, 5: 0.84, 1: 0.76, 9: 0.44 |
| 4 | 3: 0.98, 1: 0.96 | 3: 0.44, 1: 0.76, 2: 0.72, 5: 0.867 | 3: 0.44, 1: 0.62, 2: 0.82, 5: 0.72, 4: 0.7, 7: 0.8 |
| 5 | 9: 0.973, 1: 0.993 | 9: 0.613, 1: 0.893, 5: 0.867, 4: 0.787 | 9: 0.4, 1: 0.8, 5: 0.6, 4: 0.72, 8: 0.84, 2: 0.86 |
| 6 | 7: 0.967, 5: 0.98 | 7: 0.813, 5: 0.893, 4: 0.773, 6: 0.787 | 7: 0.82, 5: 0.84, 4: 0.74, 6: 0.9, 1: 0.88, 0: 0.82 |
| 7 | 3: 0.913, 8: 0.933 | 3: 0.853, 8: 0.827, 7: 0.813, 9: 0.613 | 3: 0.74, 8: 0.7, 7: 0.74, 9: 0.54, 5: 0.64, 2: 0.84 |
| 8 | 7: 0.84, 9: 0.987 | 7: 0.747, 9: 0.947, 6: 0.907, 0: 0.987 | 7: 0.64, 9: 0.84, 6: 0.9, 0: 0.9, 2: 0.76, 3: 0.88 |

Table 3: Full results on Fashion MNIST. Entry formatting same as Table 2.

| Run | T1 | T2 | T3 |
|---|---|---|---|
| 1 | 6: 0.77, 4: 0.85 | 6: 0.72, 4: 0.77, 8: 0.93, 5: 0.92 | 6: 0.6, 4: 0.8, 8: 0.9, 5: 0.94, 0: 0.0, 7: 0.0 |
| 2 | 3: 0.99, 8: 0.97 | 3: 0.83, 8: 0.95, 7: 0.93, 6: 0.69 | 3: 0.88, 8: 0.84, 7: 0.86, 6: 0.32, 2: 0.74, 5: 0.82 |
| 3 | 1: 0.96, 4: 0.98 | 1: 0.83, 4: 0.64, 2: 0.67, 3: 0.75 | 1: 0.78, 4: 0.66, 2: 0.66, 3: 0.56, 7: 0.94, 8: 0.92 |
| 4 | 9: 1.0, 2: 1.0 | 9: 0.89, 2: 0.93, 8: 0.87, 7: 0.89 | 9: 0.9, 2: 0.7, 8: 0.82, 7: 0.8, 6: 0.32, 0: 0.86 |
| 5 | 7: 1.0, 6: 1.0 | 7: 0.92, 6: 0.92, 3: 0.71, 9: 0.93 | 7: 0.72, 6: 0.7, 3: 0.68, 9: 0.9, 1: 0.82, 5: 0.66 |
| 6 | 0: 0.93, 8: 0.95 | 0: 0.91, 8: 0.96, 4: 0.77, 2: 0.6 | 0: 0.84, 8: 0.94, 4: 0.78, 2: 0.48, 5: 0.84, 9: 0.92 |
| 7 | 8: 0.99, 4: 0.98 | 8: 0.91, 4: 0.79, 6: 0.55, 1: 0.95 | 8: 0.92, 4: 0.76, 6: 0.42, 1: 0.9, 7: 0.74, 0: 0.68 |
| 8 | 8: 0.97, 0: 0.95 | 8: 0.92, 0: 0.99, 7: 0.92, 9: 0.73 | 8: 0.9, 0: 0.64, 7: 0.9, 9: 0.88, 6: 0.66, 1: 0.82 |

[1ex] height

Table 4: Mean class accuracies on different $T_{CP}$ values on MNIST. CX represents digit/class X. For there experiments, LR=2.

| $T_{CP}$ | C0 | C1 | C2 | C3 | C4 | C5 | C6 | C7 | C8 | C9 |
|---|---|---|---|---|---|---|---|---|---|---|
| 0.05 | 0.81 | 0.53 | 0.79 | 0.54 | 0.93 | 0.52 | 0.69 | 0.75 | 0.59 | 0.73 |
| 0.10 | 0.89 | 0.91 | 0.76 | 0.6 | 0.65 | 0.67 | 0.78 | 0.72 | 0.39 | 0.67 |
| 0.15 | 0.68 | 0.62 | 0.57 | 0.74 | 0.54 | 0.61 | 0.87 | 0.69 | 0.54 | 0.77 |
| 0.20 | 0.78 | 0.76 | 0.74 | 0.65 | 0.66 | 0.78 | 0.64 | 0.82 | 0.49 | 0.55 |
| Avg. | 0.79 | 0.73 | 0.73 | 0.62 | 0.65 | 0.65 | 0.75 | 0.73 | 0.5 | 0.67 |

A.6   INFLUENCE OF $T_{CP}$ PARAMETER

Table 5 show how the performance changes with $T_{CP}$ value. We see that there is considerable increase in number of non-detected tasks as $T_{CP}$ is increased, but the net performance after three tasks is around the same range.[13]

---

[13]Note that in comparison to experiments in the rest of the paper, these runs were conducted with a learning rate of 2, which is the main reason why performance falls short compared to what is observed in the main text.

Table 5: Average test accuracies (8 runs) for each $T_{CP}$ value. Entries under TX represent the net accuracy of the network in the classification of all classes that belonged to the tasks including and before index X (e.g. T2 is the accuracy on four classes, two from task 1 and 2 each). Number of runs with non-detected tasks is provided under entry "ND". All experiments are on MNIST.

| Task ND | $T_{CP}$ | T1 | T2 | T3 |
|---|---|---|---|---|
| $T_{CP} = 0.05$ | 0.92 | 0.80 | 0.67 | 1/8 |
| $T_{CP} = 0.10$ | 0.91 | 0.72 | 0.71 | 2/8 |
| $T_{CP} = 0.15$ | 0.89 | 0.72 | 0.67 | 3/8 |
| $T_{CP} = 0.20$ | 0.85 | 0.77 | 0.69 | 3/8 |
| [1ex] height | | | | |

