# OpenReview forum: "Directed Structural Adaptation to Overcome Statistical Conflicts and Enable Continual Learning"
_ICLR.cc/2025/Conference — Submitted to ICLR 2025_

### Official Review · Reviewer_KzWv · 2024-10-27

**Soundness:** 1
**Presentation:** 1
**Contribution:** 1
**Rating:** 1
**Confidence:** 5

**Summary:**

This paper claims to design a continual learning approach that can non-parametrically scale with the complexity of the data. They show their performance on two grayscale vision datasets: MNIST and FMNIST.

**Strengths:**

I did not find any strength in the paper.

**Weaknesses:**

The paper is hard to follow and uses technical terms and math notations without first defining them and they use confusing terminology without properly setting the stage for the discussion. The figures are vague and small, captions are inadequate to describe the experiments done.

**Questions:**

1. The paper only considers average accuracy which is a poor indicator of performance of a continual learning (CL) system. In a CL system, there are multiple tasks in the environment which arrive sequentially. The dynamics between the task performance is important to characterize the system's capability of transferring knowledge between tasks. Reporting only accuracy may mask the performance in other tasks. For example, consider a CL system with 3 tasks having accuracies 100 (task 3), 70 (task 2) and 10 % (task 1). The average accuracy will be 60% which completely masks the poor performance on task 1. However, as can be seen, the system drastically forgets the older tasks and focuses on the current task more. This is a general phenomenon for most of the learners and CL aims to avoid that drastic reduction in older task accuracy (popularly known as catastrophic forgetting). So reporting transfer as well as forward and backward transfer is necessary to fully characterize the system. Another important feature to consider in a CL system is its memory overhead. These CL systems are meant to perform in an environment with a lot of tasks and how they grow and perform with the task complexity is important. See references 1, 2 for other performance statistics like transfer, forward and backward transfer, memory overhead. Moreover, they chose EWC as a baseline which is a really old method and I believe it requires task ids which is different from their setup. However, I never found EWC performing as poor as the author reported.

2. The proposed approach does not address catastrophic forgetting, In Table 1 their accuracy degrades over tasks and this indicates they may completely forget the older tasks if they had tested their system on more tasks. To eliminate this criticism, an experiment with a lot of tasks from imagenet would help. They choose poor baselines. There are methods in the literature that can improve on past tasks by virtue of seeing new task (backward transfer). A comparison with such approaches in [1,3,4] would help to quantify their novelty. In CL, the goal is to incrementally improve the model representation while having access to more data from different tasks. If the model breaks the plasticity-rigidity trade-offs and overfits to the recent tasks, the representation does not improve over time. I think the field has moved beyond trying to avoid forgetting [1,3], rather the goal should be to improve performance on older as well as recent tasks.

3. They consider only 3 tasks and chose grayscale datasets. To my experience, MNIST does not require even CNNs to perform well and can perform really well even with an MLP. It is popularly known that these datasets are of lower complexity. Methods that perform well on MNIST may break when tested on more complex RGB datasets like CIFAR10, CIFAR100, Imgaenet, FOOD1k. As the authors do not provide any theoretical proof that their CL can retain performance on a large enough task-sets and relies heavily on empirical validation, it is useful to validate their model over large and varied datasets. Otherwise, they may be overfitting to a particular simple scenario. CL systems are not useful unless they can perform on a lot of tasks (~100-1000 tasks) on a relatively constrained resource setting.

4. They lack a detailed literature review. The field is vast and diverse. Some methods use constant capacity (EWC, O-EWC, LwF) while others grow capacity [1,2,3] as more tasks arrive. Even there are different types of replay strategies in the literature which rehearse on older task data to mitigate forgetting. Without comparing and contrasting their approach with different groups in the literature, it is really hard to make decision on their performance.

5. The biggest flaw I found is lack of detailed experiments and poor writing. It is hard to follow their flow of writing and they abruptly start describing their model without any pre-context. CL systems are complicated. There are multiple algorithmic modifications contributing to a particular feature. Hence, for any CL system, it is necessary to do ablation study to tweak out the relative contribution of different algorithmic modification in comparison with the control group. Another important experiment is adversarial experiment where one tries to break the algorithm by providing adversarial tasks or making the environment really complex. These experiments provide valuable insights on how the model behaves and when it may fail. The paper does not include any of the above experiments.

[1] https://arxiv.org/abs/2004.12908
[2] https://arxiv.org/pdf/2012.12631
[3] Ramesh, Rahul, and Pratik Chaudhari. "Model zoo: A growing" brain" that learns continually." arXiv preprint arXiv:2106.03027 (2021).
[4] Ruvolo, Paul, and Eric Eaton. "ELLA: An efficient lifelong learning algorithm." International conference on machine learning. PMLR, 2013.

---

> ### Author Response · Authors · 2024-11-12
> **Response**
>
> Despite their confidently-reactive attitude towards the work, the reviewer seems to have not understood or overlooked important parts of the paper, as detailed below in our responses to their questions:
>
> 1) Tables 2 and 3 in Appendix A.5 (unnecessarily overcrowded for the main text compared to their compact representation in Table 1) display the decomposition of accuracies per task, and per run. It is clear that across different runs, accuracies per task (after training on all tasks) are generally evenly distributed, except for one case when a task was not detected (Fashion MN, run 1). Regarding EWC, we train it without explicit early stopping, which is probably the reason why it eventually forgets completely. This is a more realistic scenario as a continual learning system by definition is to operate with an unbounded stream of data. It might be overlooked because they are not references explicitly in the main text (despite Appendix A.5 was), this explicit reference can be included in the final version.
>
> 2,4) As far as I see the references that the reviewer gives as example methods in literature all use either replay or externally defined task IDs, hence are not comparable to our method (which requires neither, as we state multiple times in the paper).
>
> 5) Our continual learning method does not have any decomposable mechanism that can be subjected to ablation experiment. If the reviewer can see different mechanisms that can be subjected to ablation study that we don't see ourselves, they are welcome to suggest it.
>
> The other criticisms of the reviewer, which can be summarized as the insufficiency of baselines and complexity of the method, are already discussed and acknowledged in the paper, within the context of this method being an proof-of-principle demonstration of a capability that is qualitatively different from what is possible with existing methods (see our response to (2,4)) and therefore, practical considerations that are needed for scaling it up to more complex experiments have not been fleshed out yet. This is openly stated in the paper.

---

> > ### Comment · Reviewer_KzWv · 2024-11-12
> >
> > The authors put the least effort to reflect upon my feedback and instead, they took a defensive stand. They mention a table in the appendix which they do not refer in the main text and claim the accuracy is evenly distributed (which is subjective, in my opinion they are not). They vaguely refer to 'early stopping' as a plausible reason for poor performance of EWC whereas there are other hyperparatmeters and they do not have any study proving their claim. Ablation studies are common for continual learning papers and takes a lot of effort to carefully design one. The authors seem to completely ignore my suggestion for adversarial study. Overall, I find their rebuttal shallow and lacking careful consideration. Hence, I am keeping my score.

---

> ### Author Response · Authors · 2024-11-12
>
> Contrary to what the reviewer states, we are not taking a defensive stance but only pointing out the incongruence of the reviewer's comments with what is written on the paper, as it is our right to do so, and as we hope that the chairs will also take into account when they make a complete evaluation.
>
> In contrast to the reviewer's surprising claim that "even distribution is subjective", it is actually a quantifiable property. In fact, as the reviewer can also verify with a few lines of code, the mean accuracies for first, second, and third task in MNIST are 0.706, 0.747, 0.746 with reasonably low variances 0.0173, 0.0140, 0.0151. Same goes for Fashion MNIST, with means 0.798, 0.711, 0.767 and variances 0.007, 0.035, 0.021 excluding the nondetected case. Assuming we agree that an accuracy difference within the range of 5% is a reasonable number in the range of 100% (and given that the existing variance is not also biased towards being higher in earlier tasks), we can objectively say that they are evenly distributed.
>
> Likewise, the reviewer still doesn't recommend a concrete ablation study or address our concern that there seems to be no appropriate decomposition of our method to be subjected to that, instead only saying that they are "common for continual learning papers."
>
> Our EWC parameters are in detailed experimental setup in the appendix and are a standard choice for the algorithm. Continued learning of EWC is a reasonable explanation for eventual failure since EWC acts by regularization, not by explicit protection of information by isolating parts of the network. In any case EWC works by task boundaries and hence is not central to our narrative in the paper. Their proposal of adverserial study falls into our consideration of more complex experiments in future iterations (last paragraph of the rebuttal), and is considered within it. The reviewer seems to not object that the references he sent as example of other methods are not comparable to ours' as they require replay or task boundaries, so we guess we are agree on that part.
>
> Nonetheless we thank you for your comments.

---

> ### Comment · Reviewer_KzWv · 2024-11-15
>
> I have several comments:
> 1. I do not see those numbers, let alone the extremely low variances  in Appendix A5 as the author mentioned in his comment.
>
> 2. Ablation study means ablating model's different aspects to see the effect. For example, they grow nodes, edges and one ablation study could be what happens if they stop growing their network, another could be what if they constrain the number of sample per task the model gets.
>
> 3. Could the authors provide reference to their claim that the EWC training procedure they use is standard?
>
> 4. If all the core features of continual learning goes into future work, then I believe the paper requires more work to be a continual learning paper.
>
> 5. I would say all other approaches which are task-aware have backward transfer. I wonder whether they can achieve the same performance if they use the additional information available in the task id. Ignoring a vast literature and only focusing on a narrow set of approaches in the literature review and experiments make the work weak.

---

### Official Review · Reviewer_cGue · 2024-11-01

**Soundness:** 3
**Presentation:** 2
**Contribution:** 2
**Rating:** 3
**Confidence:** 3

**Summary:**

Seeing that neural networks are prone to catastrophic forgetting due to their fixed topologies, this work proposes a new method of structural adaptation, namely DIRAD, which leads to a new type of networks that grow with a minimal complexity for a single task learning. Based on this adaptation method, this paper further proposes a framework so-called PREVAL to prevent forgetting (destructive adaptation or DA as defined by the authors in the paper) in continual learning (CL), through a two-stage learning and new task detection with L0 and L1 networks. Experiments on MNIST and F-MNIST are conducted to compare the performance of PREVAL with several CL approaches in neural networks in terms of the classification accuracy.

**Strengths:**

1. The proposed structural adaptation method is interesting and novel.

2. The resulted networks show less model complexity compared to neural networks.

3. DIRAD is further applied in CL as an architecture based method to address forgetting.

**Weaknesses:**

1. The major weakness lies in the scalability of the proposed method, as also pointed out in the discussion in section 6. The computational complexity of the proposed method increases a lot with high dimensional data, where the situation can be even worse in CL due to the need to handle a sequence of tasks. While the authors expect future "AI hardware" to support the proposed network structure, it is not clear how this type of networks can be used in practice and how significant the improvement is for them to replace neural networks.

2. I understand that the scale of experiments is largely limited by the computation complexity. Still, to convince the readers about the performance improvement of the proposed method, more experimental results with more datasets such as CIFAR and more advanced baselines are needed.

3. One important property of the proposed method for CL is about new task detection without data replay. However, many out-of-distribution detection techniques indeed can be incorporated with current CL techniques to detect data distribution shifts based on new task data only, e.g., a suddenly increased loss of the current model with new data most time will detect the data distribution shift. To better justify the advantage of the proposed node validation based technique, experimental study should also be conducted to compare with CL techniques with simple data distribution shift detection strategies.

4. The writing also needs to be improved. It is not easy to follow and understand what the authors are trying to say in general, particularly in section 3 and 4, e.g., how the APs are defined, how to check the conditions for edge generation.

**Questions:**

1. What is the relationship between the immediate AP and the total AP? This is not clear as least from the main body.

2. In Figure 1b, why the immediate AP of the node is exhausted but its total AP is not? Why does the generated edge have a zero gradient?

3. A priority ordering mechanism is introduced to grow the network and the authors took a conservation scheme by only performance a single generative process per step. This seems still not to answer the question "when to stop growing" (or "how much growth is necessary").

4. It seems that PREVAL needs to store multiple models and evaluate new data on all of these models. This will increase the complexity a lot in terms of both memory size and computation cost.

5. While I agree that using neural networks to predict internal nodes can cost much more compared to DIRAD, it is not clear if doing this thing, i.e., predicting internal nodes, is necessary or worth at the first place.

6. How is the inference conducted? As shown in Figure 2, the test sample is processed by the best-matching model. But during inference no task ID or ground truth is provided, how to determine the best matching model?

---

> ### Author Response · Authors · 2024-11-17
>
> We thank you for your comments and will take them into consideration in future iterations of the writing.
> Our responses to your questions:
>
> 1, 2) Immediate AP is the sum of delta (cost gradient) terms per batch on a given edge. Total AP is the sum of the magnitudes on delta terms. Distinction is that immediate AP can be zero (meaning adaptation has stuck, i.e. local optimum in NNs) while total AP can be very high (meaning there is still a large potential for learning there given correct modulation of signs of terms can be made, as we do with ENC). Figure 1b represents such a case: You have two large positive deltas, two large negative deltas (since the initial output is 0 for all cases but targets of y are +, -, -, +) resulting in a net zero delta and hence net zero gradient. In 1c, this is resolved by edge-node conversion and the newly-generated edge (x0-h) has a large gradient formed by all positive deltas.
>
> 3) This is mentioned in implementation details on A.2.2, it's not related to the priority ordering algorithm. If the mismatch in an output is <0.01, we regard the gradient from this output as 0, effectively having no effect on learning anymore and stopping growth related to itself. We omitted this from main text due to spatial constraints, but perhaps a short explicit reference would be in order.
>
> 4) Yes, although growing on top of existing networks can in principle be done thanks to the growth mechanism, we do not explore that in this version.
>
> 5) There is no guarantee that detectable conflicts will manifest in the input, they may be detectable only in internal representations. For limited capability detecting inputs may suffice, but no guarantee of its working in general. Our method makes use of all available system representations for that. We wanted to develop a concept for the most general case in this work.
>
> 6) In lines 295-302, in particular: "If there is a model for which batch is validated, that model (or the one with least ratio of invalid samples if there are multiple) is chosen to process it." Inference is no different, the only difference is that there we do not require validation (since we do not create new models during testing) and directly take the model with least ratio of invalid samples.

---

### Official Review · Reviewer_NNb9 · 2024-11-05

**Soundness:** 2
**Presentation:** 1
**Contribution:** 2
**Rating:** 3
**Confidence:** 3

**Summary:**

This paper proposes a novel technique to train adaptive networks by greedily adding nodes to a network when the edge weights exhibit zero gradient after training on a batch (saturation). The intended goal is to handle catastrophic forgetting in continual learning settings by gradually incrementing the network capacity when the capacity is full. A mechanism that requires training another network sharing the same set of nodes can be utilized to perform task detection for CL setting (see Sec 4). The method is benchmarked on MNIST and FashionMNIST and performs better than two baselines (EWC, MAS) and consistently worse than the other (PNN+FAE).

**Strengths:**

The idea seems to be completely original, with multiple novel components and procedures.

**Weaknesses:**

The proposed method is completely new and extremely complex. Section 3 and 4 are incapable of explaining the methodology properly. Many terminologies are coined, yet the relevance of coining these terminologies are ill-justified. The complexity of the method requires many design choices, yet these choices see to be determined arbitrarily (see P16L819 "Among the many ways to realize this condition, we choose to let .. Again, among infinite alternatives, we decide to set bk,1 = 0 and σ1 to be a scaled and shifted logistic function"). I even read the full Appendix yet failed to capture the essence of the proposed method.

The paper is poorly written. Pieces of information are scattered all around. Terminologies are not clearly defined before usage. Run-on sentences make the arguments hard to parse (e.g. P15L772 "Our goal with this point of modulation is to modulate the opposing yet nonzero gradients that the original edge was once under the influence of, in a manner that aligns them so that their adaptive potential, as quantified by their total magnitude, can be fully utilized without falling for statistical stable points.").

The main concern I had was, is designing such framework necessary when plugging in a familiarity autoencoder into an adaptive network sufficient to achieve the intended goal of performing adaptive increasing of network capacity while dealing with different tasks in continual learning? What is the justification for taking such complex routes yet yielding such marginal gains empirically and doesn't scale to normal sized models and datasets?

**Questions:**

* The terminology "statistical conflict" is not formally defined, despite it being a key motivation for designing DIRAD.
* Couldn't the familiarity autoencoder be plugged into any adaptive networks to support continual learning with automatic detection of new data and assigning encountered data to suitable models adapted to process them?
* There are many self-expandable / adaptive network papers mentioned in the related works session. Why are none of those part of the baseline for comparison?
* How does the proposed technique contextualize with respect to the grand scheme of existing adaptive networks?
* How does the proposed technique generalizes its performance to hold out dataset? The greedy approach of ENC is prone to either being stuck in local minima or overfitting to noise in the training data, since whenever the edge saturation condition triggers, the model expands its capacity to accommodate the training batch.

---

> ### Author Response · Authors · 2024-11-17
>
> We thank you for your comments and will take them into consideration in future iterations of the writing.
> "Couldn't the familiarity autoencoder be plugged into any adaptive networks to support continual learning with automatic detection of new data and assigning encountered data to suitable models adapted to process them?"
> Familiarity AE can only back-predict the input, not the network. There is no guarantee that detectable conflicts will manifest in the input, they be detectable only in internal representations. For a limited continual learning capability FAEs can be used for the same purpose (we in fact do this in another study of ours) but for this one we wanted to keep it as general as possible since we are interested more in introducing the concept.
>
> "There are many self-expandable / adaptive network papers mentioned in the related works session. Why are none of those part of the baseline for comparison?"
> For various reasons as explained in that related work section. To summarize: (1) They have no guarantee of minimality or optimality as DIRAD does. (2) They often rely on an external loop (usually a genetic algorithm - if they are not "grown" with a genetic algorithm in the first place) in addition to an inner loop that realizes parameter learning; increasing computation requirements by at least an order of magnitude compared to ours, and (3) as far as we know they have not been applied to continual learning without replay or task boundaries.
>
> "How does the proposed technique contextualize with respect to the grand scheme of existing adaptive networks?"
> We think it is clear in introductory sections of the paper, some elaboration of the question would be appreciated.
>
> "How does the proposed technique generalizes its performance to hold out dataset? The greedy approach of ENC is prone to either being stuck in local minima or overfitting to noise in the training data, since whenever the edge saturation condition triggers, the model expands its capacity to accommodate the training batch."
> Our accuracy reports in Results section are already on test set (we assume that's what is meant by hold out dataset).

---

> > ### Comment · Reviewer_NNb9 · 2024-11-17
> >
> > > Familiarity AE can only back-predict the input, not the network.
> >
> > Define "back-predict". What is the difference between "back-predicting" the input vs network?
> >
> > > There is no guarantee that detectable conflicts will manifest in the input, they be detectable only in internal representations.
> >
> > Internal representations are mapped from the input with a deterministic function. Autoencoders are expected to encode the compressed representation and thus anomaly in the internal representation is expected to be reflected on the input.
> >
> > > For a limited continual learning capability FAEs can be used for the same purpose (we in fact do this in another study of ours) but for this one we wanted to keep it as general as possible since we are interested more in introducing the concept.
> >
> > Define "limited CL capability". What hard constraints do FAEs introduce?
> >
> > > (1) They have no guarantee of minimality or optimality as DIRAD does. (2) They often rely on an external loop (usually a genetic algorithm - if they are not "grown" with a genetic algorithm in the first place) in addition to an inner loop that realizes parameter learning; increasing computation requirements by at least an order of magnitude compared to ours, and (3) as far as we know they have not been applied to continual learning without replay or task boundaries.
> >
> > These points do not justify not comparing them. It's interesting how the authors claim that the computation requirement of other baselines is high whereas their own method has significant scaling issues even on normal sized datasets / models.
> >
> > > We think it is clear in introductory sections of the paper, some elaboration of the question would be appreciated.
> >
> > Specifically the components of the method explained in this work is not connected to existing works in any way. Merely mentioning related works is insufficient. How are existing works similar to this method here? Unless the authors wish to claim that their method is *completely* novel, in which it can only be judged from the empirical performance because there is no way to contextualize it with respect to existing works in the methodology.
> >
> > > Our accuracy reports in Results section are already on test set (we assume that's what is meant by hold out dataset).
> >
> > The consensus amongst all reviewers is that MNIST is too simple of a task to actually indicate any realistic properties of the method. Can you directly comment on the potential of overfitting given the greedy mechanism of the method? An alternative would be to introduce label noise to the training set of MNIST and see how the generalization to holdout set holds under noisy settings.
> >
> > The most important question "What is the justification for taking such complex routes yet yielding such marginal gains empirically and doesn't scale to normal sized models and datasets?" remains unanswered.

---

### Meta-Review · Area_Chair_MHgq · 2024-12-21

**Metareview:**

The paper introduces DIRAD, a novel method for structural adaptation that claims to enable networks to expand with minimal complexity when learning a single task. However, the presentation of the paper is quite poor and the paper is hard to follow; e.g., terminologies are not clearly defined before usage and sentences are very long which hinders the clear understanding of the main idea. Moreover, the method is evaluated with only using simple, grayscale images and more justification for its usefulness is definitely needed for accepting the paper. Thus, for the current state, the decision is Reject.

**Additional Comments On Reviewer Discussion:**

The reviewers unanimously argued about the poor writing and presentation of the paper -- they all said the writing is poor and it was very hard to follow the main idea of the paper. Moreover, as the reviewer KzWv has pointed out, the experimental results were not enough to convincingly show the effectiveness of their method. Authors provided some rebuttal, but all the reviewers still remained negative about the paper.

---

### Decision · Program_Chairs · 2025-01-22

Reject